# TOWARDS ROBUST FEATURE LEARNING WITH T-VFM SIMILARITY FOR CONTINUAL LEARNING

**Bilan Gao**
Department of Artificial Intelligent
Chung-Ang University
Seoul, Korea
bilan2020@cau.ac.kr

**YoungBin Kim**
Department of Imaging Science and Arts
The Graduate School of Advanced Imaging Science,
Multimedia and Film,
Chung-Ang University
Seoul, Korea
ybkim85@cau.ac.kr

## ABSTRACT

Continual learning has been developed using standard supervised contrastive loss from the perspective of feature learning. Due to the data imbalance during the training, there are still challenges in learning better representations. In this work, we suggest using a different similarity metric instead of cosine similarity in supervised contrastive loss in order to learn more robust representations. We validate the our method on one of the image classification datasets Seq-CIFAR-10 and the results outperform recent continual learning baselines.

## 1 INTRODUCTION

Continual learning is the research direction that mainly tackles the problem of catastrophic forgetting while a model learns new data consistently. To mitigate the catastrophic forgetting phenomenon, one strategy is to enhance the learned representation while training. An effective approach for achieving this is to apply supervised contrastive learning (Cha et al., 2021; Mai et al., 2021a;b; Han & Guo, 2021; Davari et al., 2022). Neglecting the consideration of the imbalance phenomenon between current training samples and replayed samples can cause loss of intra-class information while learning representations.This is because cosine similarity within the supervised contrastive loss, can extract features with large within-class variance, resulting in degraded model performance. To enhance the stability and reliability of feature extraction, we drew inspiration from (Kobayashi, 2021), we propose to use a similarity function based on the von Mises-Fisher distribution, further extended with a student t-distribution(t-vMF) to replace cosine similarity. We adopt the method on asymmetric supervised contrastive loss(Cha et al., 2021), and conduct experiments under offline continual learning settings, and the results show effective improvement on prediction accuracy.

## 2 T-VMF SIMILARITY ADOPTED IN CONTINUAL REPRESENTATION LEARNING

An overview of asymmetric supervised contrastive loss $L_{ASC}$(Cha et al., 2021). A base encoder $f(\cdot)$ receives as input two augmented versions $\tilde{x}_i$ and $\tilde{x}_p$ from data sample $x$ with it's label $y$. To using an inner product to measure distances, this representations are further mapped to a feature space with projection model $g(\cdot)$. The final extracted features can be written as $z_i = g(f(\tilde{x}_i))$ and $z_p = g(f(\tilde{x}_p))$. Differ from (Mai et al., 2021a), $L_{ASC}$ selects anchor samples only from current training data rather than consider all the samples in the training stage. Let $S$ denotes all the indices of the current learning samples in a single batch and $P(i)$ denotes the indices of all positive samples that are distinguished from $i$. $L_{ASC}$ can be defined as follows:

$$L_{ASC} = \sum_{i \in S} \frac{-1}{|P(i)|} \sum_{p \in P(i)} log \frac{exp(sim(z_i \cdot z_p)/\tau)}{\sum_{a \in A(i)} exp(sim(z_i \cdot z_a)/\tau)}), \tag{1}$$

where $sim(z_i \cdot z_p)$, a cosine similarity function can be written as $= \|z_i\|\|z_p\|cos\theta$ with an angle $\theta$ between $z_i$ and $z_p$. And $\tau$ is a temperature hyperparameter. $P(i)$ denotes the indices of all positive

Table 1: Classification accuracy for Seq-CIFAR-10. All results are averaged over ten independent trials. Best performance are marked in bold.

| Buffer | Baselines | Class Incremental | Task Incremental |
|--------|-----------|-------------------|------------------|
| 200 | $HAL$ (Chaudhry et al., 2020) | $32.36 \pm 2.70$ | $82.51 \pm 3.20$ |
| | $DER$ (Buzzega et al., 2020) | $61.93 \pm 1.79$ | $91.40 \pm 0.92$ |
| | $DER++$ (Buzzega et al., 2020) | $64.88 \pm 1.17$ | $91.92 \pm 0.60$ |
| | $Co^2L$ (Cha et al., 2021) | $65.57 \pm 1.37$ | $93.43 \pm 0.78$ |
| | **Ours** | $\mathbf{67.98 \pm 1.29}$ | $\mathbf{95.10 \pm 0.42}$ |
| 500 | $HAL$(Chaudhry et al., 2020) | $41.79 \pm 4.46$ | $84.54 \pm 2.36$ |
| | $DER$ (Buzzega et al., 2020) | $70.51 \pm 1.67$ | $93.40 \pm 0.39$ |
| | $DER++$ (Buzzega et al., 2020) | $72.70 \pm 1.17$ | $93.88 \pm 0.60$ |
| | $Co^2L$ (Cha et al., 2021) | $\mathbf{74.26 \pm 0.77}$ | $\mathbf{95.90 \pm 0.26}$ |
| | **Ours** | $71.68 \pm 0.41$ | $95.51 \pm 0.19$ |

samples that are distinguish from i; it is equivalent to $\{p \in A(i) : \tilde{y}_p = \tilde{y}_i\}$. $|P(i)|$ is the number of elements in set $P(i)$.

We propose to use the t-vMF Similarity(Kobayashi, 2021) to replace the cosine similarity in equation1. We first build a von Mises–Fisher distribution(mar; Banerjee et al., 2005) based similarity $p(\tilde{z}_i; \tilde{z}_p, \kappa) = C_\kappa exp(\kappa cos\theta)$ between features $\tilde{z}_i$ and $\tilde{z}_p$. It produces a probability density function on $p$ dimension with with parameter $\kappa$. The parameter $\kappa$ controls concentration of the distribution, and $C_\kappa$ is a normalization constant. Apply a t-distributed profile function $f_t(d; \kappa) = \frac{1}{1+\frac{1}{2}\kappa d^2}$ on $p(\tilde{z}_i; \tilde{z}_p, \kappa)$, the equation of t-vFM similarity function between two features $\tilde{z}_i$ and $\tilde{z}_p$ is formalized as follows(More detail in Appendix A.1):

$$\phi_t(cos\theta; \kappa) == \frac{1 + cos\theta}{1 + \kappa(1 - cos\theta)} - 1 \qquad (2)$$

Adopt equation2 in the original $L_{ASC}$, the loss function of our proposed method is shown as follows:

$$L_{Ours} = \sum_{i \in S} \frac{-1}{|P(i)|} \sum_{p \in P(i)} log \frac{exp(\phi_t(cos\theta; \kappa)/\tau)}{\sum_{a \in A(i)} exp(\phi_t(cos\theta; \kappa)/\tau)}). \qquad (3)$$

The larger value $\kappa$ has, that smaller compact region between two representation is.

## 3 EXPERIMENT AND DISCUSSION

Following prior work (Cha et al., 2021), we demonstrate the our method by training ResNet-18 (He et al., 2016) as a base encoder on Seq-CIFAR-10 dataset (Krizhevsky et al., 2009). We validate our method on class incremental learning and task incremental learning scenarios with replay buffers of size 200 and 500 (Buzzega et al., 2020).The value of $\kappa$ in Equation 3 is set to 16.(More detail in Appendix A.2). As shown in Table 1, our proposed method outperforms all the baselines with buffer 200, which indicates that the t-vMF similarity shows promise. In the case of buffer size 500, even though the proposed method does not outperform all baselines, it approaches the best baselines.

## 4 CONCLUSION AND FUTURE WORK

In this work, we delve deeply into continual representation learning, using t-vMF similarity to enhance feature reliability and outperform most baselines in terms of prediction accuracy. Our future aim is to further investigate alternative representation learning approaches and various metrics to improve continual representation learning.

ACKNOWLEDGEMENTS

This research was supported by Basic Science Research Program through the National Research Foundation of Korea(NRF) funded by the Ministry of Education(NRF-2022R1C1C1008534), and Institute for Information & communications Technology Planning & Evaluation (IITP) through the Korea government (MSIT) under Grant No. 2021-0-01341 (Artificial Intelligence Graduate School Program, Chung-Ang University).

URM STATEMENT

Author Bilan Gao meets the URM criteria of ICLR 2023 Tiny Papers Track.

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

## A APPENDIX

### A.1 MORE DETAIL EXPLANATION ON T-vFM SIMILARITY

T-vFM similarity has the foundation on a similarity function that based von Mises–Fisher distribution(mar; Banerjee et al., 2005), a von Mises–Fisher distributed similarity which produces a probability density function on $p$ dimension with parameter $\kappa$ between two features $\tilde{z}_i$ and $\tilde{z}_p$ in $d$ dimension can be formulated as:

$$p(\tilde{z}_i; \tilde{z}_p, \kappa) = C_\kappa exp(\kappa \tilde{z}_p^\top \tilde{z}_i) \tag{4}$$
$$= C_\kappa exp(\kappa cos\theta). \tag{5}$$

The parameter $\kappa$ controls concentration of the distribution, and $C_\kappa$ is a normalization constant. To find out the part that mainly support the distribution $p(\tilde{z}_i; \tilde{z}_p, \kappa)$, define a profile function $f_e$ with $d$ and $\kappa$, then the profile function $f_e(d; \kappa)$ is defined as

$$f_e(d; \kappa) = exp(-\frac{1}{2}\kappa d^2). \tag{6}$$

Thus, Equation 5 can be reformulated by using a profile function(Equation 6) as follows

$$p(\tilde{z}_i; \tilde{z}_p, \kappa) = C_\kappa exp(\kappa - \frac{1}{2}\kappa \|\tilde{z}_i - \tilde{z}_p\|^2) \tag{7}$$

$$= C_\kappa^{'} f_e(\|\tilde{z}_i - \tilde{z}_p\|; \kappa). \tag{8}$$

Observing from Equation 8, the vMF similarity is primarily distinguished by $f_e(\|\tilde{z}_i - \tilde{z}_p\|; \kappa)$, which means the vMF similarity can formally be re-defined as an alternative for $cos\theta$ as

$$\phi_e(cos\theta; \kappa) = 2\frac{f_e(|\tilde{z}_i - \tilde{z}_p\|; \kappa) - f_e(2; \kappa)}{f_e(0; \kappa) - f_e(2; \kappa)} - 1 \tag{9}$$

$$= 2\frac{exp(\kappa cos\theta) - exp(-\kappa)}{exp(\kappa) - exp(-\kappa)} - 1 \in [1, -1]. \tag{10}$$

Equation 10 demonstrates von Mises–Fisher distributed similarity between two features $\tilde{z}_i$ and $\tilde{z}_p$ with an important parameter $\kappa$. According to its function that introduced above, changing different value of $\kappa$ can dynamically balance the trade-off between learning discriminative features for each task and minimizing interference between new task samples and learned task samples.

A previous work Van der Maaten & Hinton (2008) suggested using a heavy tail distribution to better capture distinguishing features. Modify the profile function(Equation 6) with a heavy-tailed student T distribution, the new profile function can be updated as follows:

$$f_t(d; \kappa) = \frac{1}{1 + \frac{1}{2}\kappa d^2}. \tag{11}$$

Due to the property of t-distribution, it makes the similarity function less sensitive to small changes in the learned representations. We reformulate vMF similarity with Equation 10, the formal definition of t-vFM similarity between features $\tilde{z}_i$ and $\tilde{z}_p$ is given as follows:

$$\phi_t(cos\theta; \kappa) = 2\frac{f_t(|\tilde{z}_i - \tilde{z}_p\|; \kappa) - f_t(2; \kappa)}{f_t(0; \kappa) - f_t(2; \kappa)} - 1, \tag{12}$$

$$= 2\frac{\frac{1}{1+\kappa(1-cos\theta) - \frac{1}{1+2\kappa}}}{1 - \frac{1}{1+2\kappa}} - 1, \tag{13}$$

$$= \frac{1 + cos\theta}{1 + \kappa(1 - cos\theta)} - 1. \tag{14}$$

As shown in Equation 14, t-vFM similarity can be simply calculated with $cos\theta$ and $\kappa$. Since $cos\theta \in [-1, +1]$, $f_e(\|\tilde{z}_i - \tilde{z}_p\|; \kappa)$ is resized to be adaptable with $cos\theta$. Applying Equation 14 to asymmetric contrastive loss, our proposed loss can be modified from Equation 1 as follows:

$$L_{Ours} = \sum_{i \in S} \frac{-1}{|P(i)|} \sum_{p \in P(i)} log \frac{exp(\phi_t(cos\theta; \kappa)/\tau)}{\sum_{a \in A(i)} exp(\phi_t(cos\theta; \kappa)/\tau)}). \tag{15}$$

## A.2 EXPERIMENTAL DETAILS

**Dataset and architecture**  In the training stage, we split the CIFAR-10 dataset into five distinct sets of samples,with each set consisting of two different classes. For architecture, we use a non-pretrained ResNet-18 as a base encoder for extracting features followed by a 2-layer projection MLP to map these representations to a latent space of 128 dimensions (Khosla et al., 2020).

**Hyperparameters**  We train the base encoder with a 512 batch size, and 0.5 as learning rate. The temperature hyperparameter($\tau$) for our loss is set to 0.5.

Table 2: Classification accuracy for Seq-CIFAR-10. All results are averaged over ten independent trials. Best performance are marked in bold. Our proposed method choose three values of $\kappa$: 4,16,32

| Buffer | Baselines | Class Incremental | Task Incremental |
|--------|-----------|-------------------|------------------|
| 200 | $HAL$ (Chaudhry et al., 2020) | $32.36 \pm 2.70$ | $82.51 \pm 3.20$ |
| | $DER$ (Buzzega et al., 2020) | $61.93 \pm 1.79$ | $91.40 \pm 0.92$ |
| | $DER++$ (Buzzega et al., 2020) | $64.88 \pm 1.17$ | $91.92 \pm 0.60$ |
| | $Co^2L$ (Cha et al., 2021) | $65.57 \pm 1.37$ | $93.43 \pm 0.78$ |
| | $Ours(\kappa = 4)$ | $67.94 \pm 1.04$ | $\mathbf{95.20 \pm 0.22}$ |
| | $Ours\ (\kappa = 16)$ | $67.98 \pm 1.29$ | $95.10 \pm 0.42$ |
| | $Ours\ (\kappa = 32)$ | $\mathbf{67.99 \pm 0.71}$ | $95.14 \pm 0.38$ |
| 500 | $HAL$(Chaudhry et al., 2020) | $41.79 \pm 4.46$ | $84.54 \pm 2.36$ |
| | $DER$ (Buzzega et al., 2020) | $70.51 \pm 1.67$ | $93.40 \pm 0.39$ |
| | $DER++$ (Buzzega et al., 2020) | $72.70 \pm 1.17$ | $93.88 \pm 0.60$ |
| | $Co^2L$ (Cha et al., 2021) | $\mathbf{74.26 \pm 0.77}$ | $\mathbf{95.90 \pm 0.26}$ |
| | $Ours\ (\kappa = 4)$ | $71.51 \pm 0.84$ | $95.37 \pm 0.18$ |
| | $Ours\ (\kappa = 16)$ | $71.68 \pm 0.48$ | $95.51 \pm 0.19$ |
| | $Ours\ (\kappa = 32)$ | $71.32 \pm 0.70$ | $95.43 \pm 0.26$ |

**Evaluation**   We follow class balance strategyCha et al. (2021) and use an additional linear classifier trained only on the final task samples and the buffer data with learned representations. All the results are reported with 10 trials.

**Additional experiment on different values of $\kappa$**   As shown in Table 2, we report results with different $\kappa$ values, and our algorithm surpasses all the baselines of recent works in the case of buffer 200 setting. In the comparison of 3 different values of $\kappa$, class incremental learning reaches the best performance when $\kappa = 32$, and task incremental learning achieves the best performance when $\kappa = 4$. Such results indicate that our method successfully learns and reduces the effect of data imbalance. Moreover, there are no big difference among all the results with different values of $\kappa$, which indicates a narrow region does not effect the performance in this case.

## A.3   RELATED WORKS

**Contrastive Continual Learning**   Looking at continual learning from the perspective of feature learning, the former can be broken down into following two main questions: How can the learning features be robust and more reliable? How can the learned features be maintained? There are many recent works that focused on apply contrastive loss to leverage more generalized presentations. Self-supervised contrastive loss(Chen et al., 2020) was adopted in these works (Gallardo et al., 2021; Zhang et al., 2020), which increased models' performance without using labels. Supervised contrastive loss(Khosla et al., 2020) was used in many recent works(Mai et al., 2021a; Cha et al., 2021; Han & Guo, 2021; Davari et al., 2022) to make samples that belong to the same class get closer to each other. Supervised contrastive loss was used in continual learning in the study presented by (Mai et al., 2021a), then feature propagation was introduced with supervised contrastive loss to tackle multiple continual learning tasks(Han & Guo, 2021). In (Davari et al., 2022), controlling feature forgetting was suggested using a linear classifier before and after a new task started. Another recent work(Cha et al., 2021) improved supervised contrastive loss for better feature learning, and it also introduced a relation distillation to maintain the learned representation. There haven't been many works that consider a better approach on calculating similarity under the data imbalance environment. In our work, we investigate a different type of similarity function that adopts in asymmetric supervised contrastive loss(Cha et al., 2021) to diminish intra-class variance.

**von Mises-Fisher Distribution** In recent years, von Mises–Fisher Distribution(mar) is one of the probability distributions on sphere that has been applied to many fields in deep learning. One of the previous works(Gopal & Yang, 2014) presented a von Mises–Fisher distribution-based clustering method, which is more effective on high-dimensional data. Different types of von Mises–Fisher distribution based clustering approaches have been adopted for different tasks such as semantic segmentation(Hwang et al., 2019) and graph neural network(Wang et al., 2023). Using von Mises–Fisher distribution on metric learning or loss functions (Hasnat et al., 2017; Zhe et al., 2019; Kobayashi, 2021) showed significant improvement in performance as well.

