# OpenReview forum: "Towards Robust Feature Learning with t-vFM Similarity for Continual Learning"
_ICLR.cc/2023/TinyPapers — Submitted to Tiny Papers @ ICLR 2023_

### Official Review · Reviewer_Kep2 · 2023-03-20

**Confidence:** 4

**Summary Of Contributions:**

This paper proposes a new method for improving the representation of data in continual learning by using a different similarity metric instead of cosine similarity in supervised contrastive loss. The authors suggest using a similarity function based on the von Mises-Fisher distribution, further extended with a student t-distribution (t-vMF) for representation learning. They validate their method on an image classification dataset and achieve promising results.

**Rating:**

Great Start (GS): a submission which meets some of the reviewing criteria but has room for improvement

**Strengths And Weaknesses:**

Strength:
- The paper proposes to use t-vMF as the similarity metric in supervised contrastive learning to improve representation learning in continual learning.
- The results of the experiments are promising and show that the proposed method outperforms some existing baselines.

Weakness:
1. Motivation for using a different t-vMF as a similarity metric. Cosine similarity has been proven effective in many areas. I wonder what is the motivation for replacing it? Do the authors find some specific weakness in the cosine similarity metric, and how does the proposed metric address this weakness?
2. Motivation for using it for continual learning. Why do authors specially choose the setting of continual learning to validate the proposed metric? Is the proposed metric specifically useful for continual learning, and if so, why?

See "Suggested Changes" for other weaknesses and potential improvements.

**Suggested Changes:**

I understand this is a tiny paper with 2 pages limit, but the authors can further clarify and extend these in either the main text or appendix to make the paper clearer:
1. Clarification of motivations. Per weakness 1, the authors should give a direct intuition of why t-vMF is needed and what are the benefits of it compared to cosine similarity. Per weakness 2, the authors should explain why using continual learning for evaluation, because if we want to validate the effectiveness and generalization of the proposed metric, we usually start with the most common setting (e.g. the most basic image classification) and then potentially extend to other more challenging settings.
2. The experimental settings of class incremental learning and task incremental. E.g., how many stages are there in total? How many classes/tasks are there in each stage? What are exactly the tasks? ...
3. More comparison experiments. As the proposed metric performs worse than Cha et al., 2021 when the buffer size is 500, the authors can consider adding more different buffer sizes to study how the proposed metric relates to the buffer size. E.g., does it work better when the buffer size is smaller?
4. Experimental details. What are the detailed experimental settings, e.g. how do the authors tune the hyper-parameters as I think it can make a big influence, given the results of different methods are similar. Also, how sensitive is the proposed method to the hyper-parameter $k$?

---

> ### Author Response · Authors · 2023-06-01
> **Thank you for your review**
>
> Thank your for your detail comments and feedback, we really appreciated. We modified our explanation of our motivation, a more detail explanations on experimental setting are added in the appendix section as well. Besides these, we added an additional table, which compares three different value of $\kappa$ in our method. There is an extended explanation on our proposed method that added in the appendix section as well. All the changes we have made are shown in the updated paper.

---

### Official Review · Reviewer_zX2u · 2023-03-26

**Confidence:** 4

**Summary Of Contributions:**

A new feature learning technique for continual learning is proposed.

**Rating:**

High Potential (HP): a submission which meets the reviewing criteria and has potential to make an impact on the field

**Strengths And Weaknesses:**

S1: Overall I think this paper is clear and easy to understand.

S2: Experimental results show that the proposed method is somehow effective.

W1: I would say the intuition behind the proposed t-mFV similarity is not very clear, I have to refer to the cited papers to understand more details on the method. I would recommend adding some figures to better illustrate the idea of the proposed similarity.

W2: It seems that the proposed method does not achieve the best results when the buffer size is larger, and given that the current similarity is just the t-vMF similarity from  (Kobayashi, 2021), I would be interested in how to further integrate the technique in the continual learning setting.

**Suggested Changes:**

C1: I would suggest to further clarify the intuition of the t-vMF similarity.

C2: I would suggest to bold only the best numbers in the comparison table, it is misleading to just bold the number from the proposed method.

---

> ### Author Response · Authors · 2023-06-01
> **Thank you for your review**
>
> Thank you for your review! We appreciated all the comments and feedback. We modified the content related to motivation of using t-vMF similarity in continual learning, and the best numbers in the comparison table are bolded. All the changes we have made are shown in the updated paper.

---

### Meta-Review · Area_Chair_FuL3 · 2023-04-06

**Recommendation:** Invite to archive
**Confidence:** 5

**Metareview:**

This paper proposes to use a different similarity measure that is t-vMF for supervised contrastive learning in the continual learning setting, the experimental results are clear to show that the propsod method is effective, yet it is raised by both reviewers that the motivation of using t-vMF similarity is not clear, and also more experiment and implemental details are needed to make the paper reproducible.
Overall, this paper needs some revisions to be CCR.


**Summary:**

The strength of this paper is that the experiments show that the method is effective, yet the main weakness of the paper is that the motivation is not clear, and more ablation studies are needed, one case both reviewers raised are the case where the buffer size is smaller.

**Reason For Not Giving A Higher Recommendation:**

The two reviewers both think the motivation of the method is lacking.


**Reason For Not Giving A Lower Recommendation:**

The experimental results are effective, and the revision needed are mangable.

---

### Decision · Program_Chairs · 2023-04-10

Invite to archive